# Functional disease architectures reveal unique biological role of transposable elements

Farhad Hormozdiari [1,2], Bryce van de Geijn[1,2], Joseph Nasser[2], Omer Weissbrod[1,2], Steven Gazal [1,2], Chelsea J.-T. Ju [3], Luke O' Connor [1,4], Margaux L.A. Hujoel [5], Jesse Engreitz [2], Fereydoun Hormozdiari[6,7] & Alkes L. Price [1,2,5]

Transposable elements (TE) comprise roughly half of the human genome. Though initially derided as junk DNA, they have been widely hypothesized to contribute to the evolution of gene regulation. However, the contribution of TE to the genetic architecture of diseases remains unknown. Here, we analyze data from 41 independent diseases and complex traits to draw three conclusions. First, TE are uniquely informative for disease heritability. Despite overall depletion for heritability (54% of SNPs, 39 ± 2% of heritability), TE explain substantially more heritability than expected based on their depletion for known functional annotations. This implies that TE acquire function in ways that differ from known functional annotations. Second, older TE contribute more to disease heritability, consistent with acquiring biological function. Third, Short Interspersed Nuclear Elements (SINE) are far more enriched for blood traits than for other traits. Our results can help elucidate the biological roles that TE play in the genetic architecture of diseases.

[1] Department of Epidemiology, Harvard T.H. Chan School of Public Health, Boston, MA 02115, USA. [2] Program in Medical and Population Genetics, Broad Institute of MIT and Harvard, Cambridge, MA, USA. [3] Department of Computer Science, University of California, Los Angeles, CA 90095, USA. [4] Program in Bioinformatics and Integrative Genomics, Harvard Graduate School of Arts and Sciences, Boston, MA, USA. [5] Department of Biostatistics, Harvard T.H. Chan School of Public Health, Boston, MA 02115, USA. [6] Department of Biochemistry and Molecular Medicine, University of California, Davis, CA 95616, USA. [7] MIND Institute and UC-Davis Genome Center, Davis, CA 95616, USA. Correspondence and requests for materials should be addressed to F.H. (email: hormozdiari@hsph.harvard.edu) or to A.L.P. (email: aprice@hsph.harvard.edu)

Transposable elements (TE), defined as DNA sequences that can insert themselves at new genomic locations, comprise roughly half of the human genome[1,2]. TE were initially viewed as parasitic elements whose presence reduced the host's fitness, and were thus derided as junk DNA; more recently, TE have been widely hypothesized to contribute to the evolution of gene regulation by providing new targets for transcription factor binding and rewiring core regulatory networks[3–18], and TE have been shown to play important roles in a growing number of disease-specific examples[19–22]. In addition, TE have been shown to exhibit excess overlap with regions of open chromatin and other functional annotations, but those studies did not analyze disease and complex trait data. Thus, our current understanding of the contribution of TE to the genetic architecture of diseases and complex traits is extremely limited.

The partition heritability method[23,24] is one of the methods used to increase our understanding about the genetic architecture of diseases and complex traits. These methods compute the heritability of each functional annotation and subsequently compute enrichment of each functional annotation to capture disease and trait heritability. However, recent heritability partitioning methods[25] only require GWAS summary statistics, which is available for most studies. Stratified LD-score regression (S-LDSC)[25] is one of the methods that can partition the heritability of diseases and complex traits for a given set of functional annotations.

Recently, S-LDSC[25] was introduced as an effective way to assess the heritability enrichment (and conditional informativeness for disease) of functional annotations by analyzing genome-wide association study (GWAS) summary statistics, which are widely available for many diseases and complex traits; in particular, S-LDSC with the baseline-LD model[26] has been shown to effectively model LD-dependent architectures. Here, we applied S-LDSC with the baseline-LD model to 41 independent diseases and complex traits (average $N = 320$ K) to estimate the components of heritability explained by different classes of TE. We sought to answer three questions. First, what is the contribution of TE to disease, and does this differ from what is expected based on the extent of their level of overlap with known functional annotations[27]? Second, do older TE contribute more to the disease heritability than younger TE? Third, do there exist classes of TE that play a greater role in specific diseases or traits?

We reached three main conclusions. First, TE are uniquely informative for disease heritability. Although TEs are only slightly depleted for disease heritability (54% of single-nucleotide polymorphisms (SNPs), $39 \pm 2\%$ of heritability where 2% denotes the standard error; enrichment of $0.72 \pm 0.03$, where 0.03 denotes the standard error; 0.38–1.23 across four main TE classes). Interestingly, they explain substantially more heritability than expected based on their depletion for regulatory and other functional annotations; this excess is concentrated outside regulatory or other functional annotations, implying that TE acquire function in ways that are currently unrecognized. Second, older TE are substantially more enriched for disease heritability than younger TE; SNPs inside the oldest 20% of TE explain 2.45× more heritability than SNPs inside the youngest 20% of TE, consistent with acquiring biological function. Third, short interspersed nuclear elements (SINE; one of the four main TE classes) are far more enriched for blood traits ($2.05 \pm 0.03$; where 0.03 denotes the standard error) than for other traits ($1.18 \pm 0.11$; where 0.11 denotes the standard error); this difference is far greater than expected based on the weaker depletion of SINEs for regulatory annotations in blood compared to other tissues

## Results

**Overview of methods.** We applied stratified S-LDSC[25] to assess the contribution of different TE to disease and trait heritability. We define a functional annotation as an assignment of a numeric value to each SNP; annotations may be binary or continuous valued (see Methods). The S-LDSC method operates by regressing chi-square association statistics on LD scores computed with respect to multiple overlapping functional annotations, and thus accounts for LD tagging effects. We estimated the heritability enrichment and standardized effect size ($\tau^\star$) for each TE annotation conditional on 75 functional annotations from the baseline-LD (version 1.1) model[26] (Supplementary Table 1, see URLs). Heritability enrichment (denoted as Observed enrichment) is defined as Observed %heritability divided by Expected (%SNPs), where Observed %heritability is the proportion of heritability causally explained by common SNPs (minor allele frequency (MAF) ≥ 0.05) in an annotation and Expected (%SNPs) is the proportion of common SNPs that lies in the annotation[25]. Distinct from Observed enrichment, we also compute the enrichment that is expected based on an annotation's overlap with baseline-LD model annotations, denoted as Expected (baseline-LD) enrichment (see Methods); this computation quantifies the extent to which heritability enrichment/depletion is explained by known functional annotations. We note that enrichment and expected enrichment can be either >1 or <1 (i.e., depletion). Standardized effect size ($\tau^\star$) is defined as the proportionate change in per-SNP heritability associated with an increase in the value of the annotation by one standard deviation[26]; unlike heritability enrichment, $\tau^\star$ quantifies effects that are unique to the focal annotation (see Methods). For each TE annotation, we include an additional annotation defined by 500 bp flanking regions, to guard against bias due to model misspecification[25] (see Methods). We have made our annotations and partitioned LD scores freely available (see URLs). Most of our results are meta-analyzed across 41 independent diseases and complex traits (average $N = 320$ K see Methods and Supplementary Data 1, same traits as in ref. [28]).

**TE are uniquely informative for disease heritability.** We first focused on four main TE classes: long interspersed nuclear elements (LINE; 21% of SNPs), SINE (16% of SNPs), long terminal repeats (LTR; 9.8% of SNPs), DNA transposons (DNA; 3.2% of SNPs), and the union all TE (ALLTE; 54% of SNPs). The proportion of SNPs in each TE class slightly exceeded the proportion of the genome spanned by the TE class (Supplementary Fig. 1). This is consistent with weaker selective constraint within surviving TE, and provides an initial indication that SNPs lying inside TE can potentially be assayed, as these SNPs have passed stringent QC filters despite the challenges of aligning TE sequences (see "Robustness of results to difficulty in mapping to TE regions" below). ALLTE explained 39% of disease heritability (meta-analyzed across 41 diseases and traits), a moderate depletion (Observed enrichment of $0.72 \pm 0.03$; Fig. 1a, b and Supplementary Table 2). The four main TE classes were all depleted or non-significantly enriched for trait heritability, with substantial heterogeneity between classes: $0.73 \pm 0.05$ for LINE, $1.18 \pm 0.11$ for SINE, $0.38 \pm 0.07$ for LTR, and $1.23 \pm 0.19$ for DNA (Fig. 1b and Supplementary Table 2). Our simulations confirm that S-LDSC produces unbiased estimates of enrichment for these annotations (see Methods, Supplementary Fig. 2). A secondary analysis of enrichment of fine-mapped causal disease SNPs[29,30] produced concordant results (Supplementary Table 3). A secondary analysis of enrichment of fine-mapped causal cis-eQTL SNPs[28] from GTEx data[31] also produced concordant results (Supplementary Table 4).

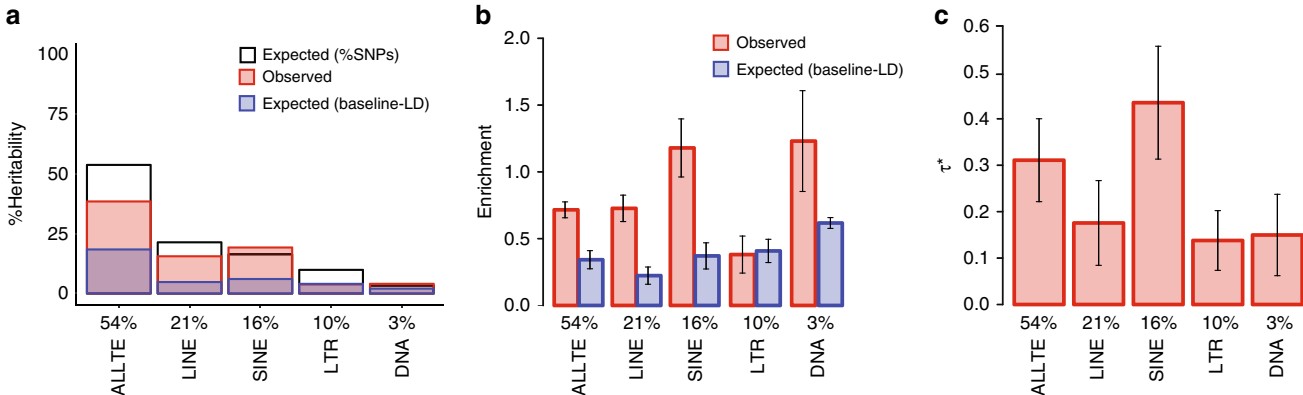

**Fig. 1** TE are uniquely informative for disease heritability. For each of four main TE classes and ALLTE, we report **a** three measures of %heritability: expected (%SNPs), observed, and expected (baseline-LD); **b** two measures of heritability enrichment: observed and expected (baseline-LD); and **c** standardized effect size ($\tau^\star$), which quantifies effect that are unique to the focal annotation. Results are meta-analyzed across 41 independent traits. Numerical values of %SNPs are provided for each annotation. Error bars denote 95% confidence intervals. Numerical results are reported in Supplementary Table 2

Notably, the heritability enrichments expected based on overlap with baseline-LD model annotations were much lower (e.g., Expected (baseline-LD) enrichment of $0.35 \pm 0.03$ for ALLTE; Fig. 1a, b and Supplementary Table 2), consistent with the large depletion of overlap between TE and known functional annotations (Supplementary Fig. 4). Thus, TE are depleted for disease heritability, but less depleted than expected based on their functional annotations (Expected (baseline-LD)), such that TE are uniquely informative for disease heritability despite being depleted for disease heritability. Accordingly, $\tau^\star$ estimates were significantly positive for each TE class (Fig. 1c), implying disease heritability enrichment effects that are not captured by known functional annotations. The $\tau^\star$ estimate of 0.43 for SINE was similar (in absolute value) to $\tau^\star$ estimates for the most informative annotations in our previous work (e.g., $-0.46$ for predicted allele age in ref. [26], 0.52 for GTEx MaxCPP in ref. [28]). Furthermore, each of the four main TE classes (LINE, SINE, LTR, and DNA) had a significant $\tau^\star$ conditional on each other and baseline-LD model annotations (Supplementary Table 5). This result indicates that each TE class is uniquely informative for disease heritability, as expected since these annotations are approximately mutually exclusive. As an additional check to verify that the significant $\tau^\star$ for all four TE classes and ALLTE would not occur due to chance, we constructed annotations of the same size based on randomly selected control regions and ran S-LDSC; as expected, we determined that enrichment estimates were not significantly different from 1 (Supplementary Fig. 3a, b) and $\tau^\star$ estimates were not significantly different from 0 (Supplementary Fig. 3c).

We investigated whether the age of a TE impacts its contribution to disease heritability. We estimated the age of each TE using miliDiv (RepeatMasker software; see URLs), which computes the number of mutations relative to a consensus sequence to estimate the age of each TE[13]. We stratified SNPs lying in a TE into five quintiles based on the age of the TE. We determined that older SNPs had higher heritability enrichments than younger SNPs (e.g., $0.91 \pm 0.11$ for oldest quintile vs $0.37 \pm 0.10$ for youngest quintile; Supplementary Fig. 5 and Supplementary Table 6). Given that both the result that TE are depleted for heritability (Observed < 1) and the result that they are less depleted than expected based on their functional annotations (Observed > Expected (baseline-LD)) were consistent across all quintiles, age alone cannot explain either of these results. We repeated this analysis for each TE class (SINE, LINE, LTR, and DNA) and observed the largest effect for SINE (Supplementary

Fig. 6). Analyses of Expected (baseline-LD) across TE families/subfamilies produced similar results, with the largest age effect for SINE (Supplementary Fig. 7 and Supplementary Data 2). These results indicate that older TE have a higher contribution to disease heritability, perhaps because they have gained biological function. Our findings are consistent with previous work reporting increased overlap with open chromatin regions for older TE[13]—although the baseline-LD model, includes a broad set of coding, conserved, regulatory and LD-related annotations.

Next, we analyzed 35 TE families/subfamilies spanning at least 0.4% of common SNPs (Supplementary Table 7). We identified 4 TE families/subfamilies that were significantly depleted for trait heritability (L1, L1PA3, ERV1, and L1PA4; Supplementary Fig. 9 and Supplementary Tables 7 and 8); none were significantly enriched. As S-LDSC is not applicable to very small annotations[25] (see Methods), for the 814 TE families/subfamilies spanning less than 0.4% of common SNPs (Supplementary Data 3), we estimated Expected (baseline-LD) enrichment only (Supplementary Data 4). We did not observe any substantial correlation between Expected (baseline-LD) enrichment and TE annotation size (Supplementary Fig. 8).

We identified 587 TE families/subfamilies that were significantly depleted for expected disease heritability (Supplementary Data 5). We also identified 46 TE families/subfamilies that were significantly enriched for expected disease heritability (Supplementary Fig. 10 and Supplementary Table 9), consistent with their excess overlap with known functional annotations (Supplementary Figs. 11 and 12 and Supplementary Datas 6 and 7). Notably, LFSINE-Vert and AmnSINE1, which have previously been reported to have important biological function[32–34], had very large expected enrichments ($5.54 \pm 0.39$ and $5.44 \pm 0.32$ respectively).

**SINE are specifically strongly enriched for blood traits.** We investigated whether TE enrichment varies across disease and traits. We estimated the heritability enrichment of each TE class (ALLTE, LINE, SINE, LTR, and DNA) for five blood traits, six autoimmune diseases, and eight brain-related traits (see Supplementary Table 10; same traits as in ref. [28]). We included a blood-specific chromatin annotation in our analyses of blood traits and autoimmune diseases, and a brain-specific chromatin annotation in our analyses of brain-related traits, in addition to the baseline-LD model (see Methods). Results are reported in Fig. 2a and Supplementary Table 11. We determined that SINE are

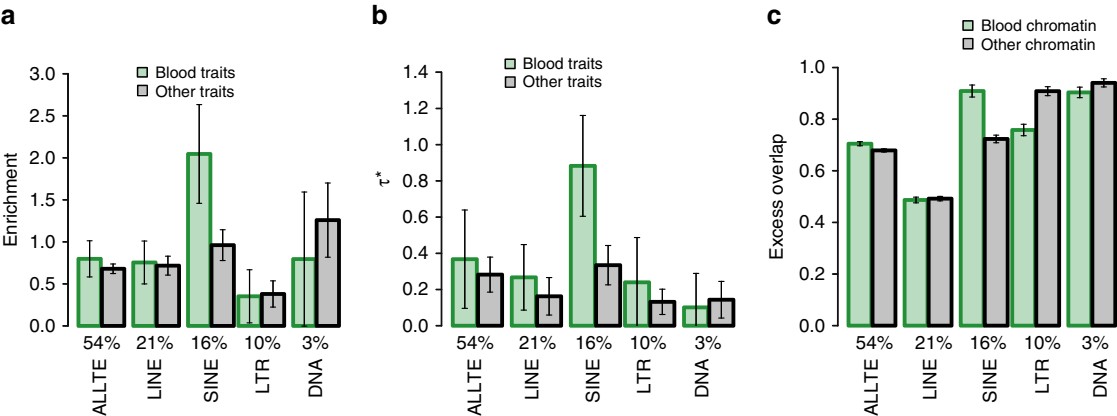

**Fig. 2** Higher SINE enrichments for blood traits. For each of four main TE classes and ALLTE, we report **a** heritability enrichment for blood traits and other traits; **b** standardized effect size ($\tau^\star$) for blood traits and other traits; and **c** excess overlap with chromatin annotations in blood and chromatin annotations in other tissues. Results are meta-analyzed across 41 independent blood and other traits. Numerical values of %SNPs are provided for each annotation. Error bars denote 95% confidence intervals. Numerical results are reported in Supplementary Tables 11a, b and 12c

specifically strongly enriched for blood traits ($2.05 \pm 0.30$ vs. $1.18 \pm 0.11$ for nonblood traits; $P = 3\text{E}{-}04$ for difference); no other TE class had significant trait class-specific enrichment after correcting for hypotheses tested, although SINE enrichment was nonsignificantly higher for autoimmune diseases vs. other traits (Supplementary Table 11). The difference in SINE enrichment for blood traits vs. nonblood traits was much higher than expected based on overlap with baseline-LD model and blood-specific chromatin annotations (Expected (baseline-LD + blood chromatin); Supplementary Table 11). Accordingly, we estimated a particularly large $\tau^\star$ for SINE for blood traits ($0.88 \pm 0.14$; Fig. 2b and Supplementary Table 11). The specific importance of SINE for blood traits is consistent with the weaker depletion of SINE in blood-specific chromatin annotations vs. other tissue/cell types (Fig. 2c and Supplementary Table 12), but is far greater than expected based on this weaker depletion; in particular, the $\tau^\star$ estimates of Fig. 2b are conditioned on blood-specific chromatin annotations.

We investigated whether motifs specific to SINE contribute to the specific importance of SINE for blood traits. For each of the 281, 9-mers occurring in the consensus sequence of the Alu family (which spans 80% of the SINE class), we defined a genomic annotation based on regions of the genome that match the 9-mer with at most one sequence mismatch (assessed using mrsFAST[35]), and analyzed the genomic annotation using S-LDSC. For one 9-mer, GCGGTGGCT, the resulting annotation (spanning 0.5% of SNPs; see Fig. 3a top panel for sequence logo) had much higher enrichment for blood traits ($6.13 \pm 1.59$) than for nonblood traits ($-0.64 \pm 0.69$; not significantly different from 0) (Fig. 3b and Supplementary Table 13); $P = 4.64\text{E}{-}05$ for difference, significant after correcting for 281 (hypotheses tested). We also determined that this genomic annotation had significantly higher excess overlap with blood open chromatin ($1.05 \pm 0.01$) compared to nonblood open chromatin ($0.86 \pm 0.01$) (Fig. 3c and Supplementary Table 13; excess overlap <1 indicates depletion), providing very strong statistical evidence that it has functionality specific to blood. We repeated the analysis for all 27 possible 9-mers with one sequence mismatch to GCGGTGGCT (effectively testing a total of $281^\star 27 = 7587$ hypotheses). For one 9-mer, GTGGTGGCT, the resulting annotation (spanning 0.6% of SNPs; see Fig. 3a bottom panel for sequence logo) had much higher enrichment for blood traits ($6.65 \pm 1.37$) than for nonblood traits ($-0.70 \pm 0.58$; not significantly different from 0) (Fig. 3b and Supplementary Table 13; $P = 4.63\text{E}{-}07$ for difference significant after correcting for $281^\star 27 = 7587$

hypotheses tested). We also determined that this genomic annotation had significantly higher excess overlap with blood open chromatin ($0.98 \pm 0.01$) compared to nonblood open chromatin ($0.81 \pm 0.01$) (Fig. 3c and Supplementary Table 13), providing very strong statistical evidence that it has functionality specific to blood. The higher excess overlap of the GCGGTGGCT and GTGGTGGCT motif annotations with blood open chromatin compared to nonblood open chromatin was observed both for motif occurrences that lie within ALU elements and for motif occurrences that do not lie within ALU elements, which have systematically higher excess overlap (Supplementary Table 14). We queried both GCGGTGGCT and GTGGTGGCT against the CIS-BP transcription factor binding motif database[36] (see URLs; we used the default CIS-BP parameters), which reported a match between GCGGTGGCT and the ZNF33A transcription factor binding motif and a match between GTGGTGGCT and the ZNF354C transcription factor binding motif. These results suggest that the specific role of GCGGTGGCT and GTGGTGGCT in blood traits may be related to ZNF33A and ZNF354C binding.

We repeated the trait class-specific analysis for the 35 TE families/subfamilies spanning at least 0.4% of common SNPs (Supplementary Tables 15–17). We did not detect any trait class-specific enrichments except for the Alu family, which spans ~80% of the SINE class and produces results similar to SINE. For the 814 TE families/subfamilies spanning less than 0.4% of common SNPs, we detected 27 that had significantly higher Expected (baseline-LD + blood chromatin) enrichment for blood-related traits vs. other traits (Supplementary Table 18; see Supplementary Fig. 13 for distribution of TE classes) and 27 that had significantly higher Expected (baseline-LD + blood chromatin) enrichment for autoimmune diseases vs. other traits (Supplementary Table 19; see Supplementary Fig. 13 for distribution of TE classes). The majority of TE families/subfamilies for that were specifically enriched for autoimmune diseases are endogenous retroviruses (ERV, which belong to LTR TE class), including MER41, which has previously been reported to contribute to autoimmune disease[16]. We also detected 109 TE families/subfamilies with higher Expected (baseline-LD + brain chromatin) enrichment for brain-related traits vs. other traits (Supplementary Data 8; see Supplementary Fig. 13 for distribution of TE classes).

**Robustness of results to mappability of TE regions.** We sought to investigate whether our results could be explained by the fact that TE regions are parts of the genome that are harder to

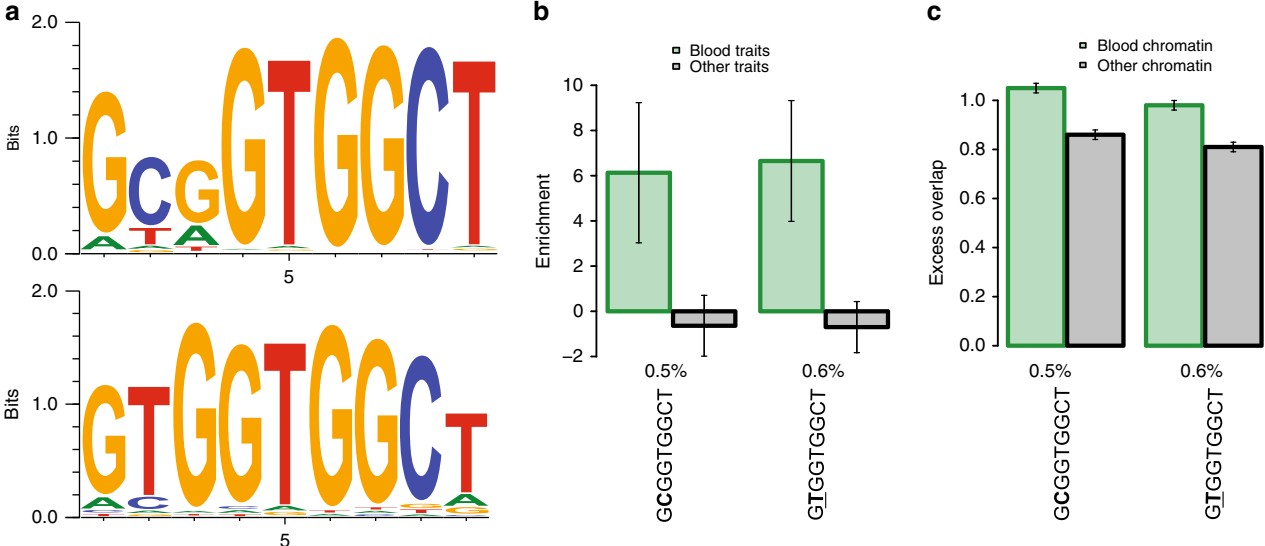

**Fig. 3** Higher enrichment of GCGGTGGCT and GTGGTGGCT for blood traits. For each of these 9-mers, we report **a** sequence logo (top panel for GCGGTGGCT and bottom panel for GTGGTGGCT); **b** heritability enrichment for blood traits and other traits; and **c** excess overlap with chromatin annotations in blood and chromatin annotations in other tissues. Results are meta-analyzed across 41 independent blood and other traits. Numerical values of %SNPs are provided for each annotation. Error bars denote 95% confidence intervals Numerical results are reported in Supplementary Table 13b, c

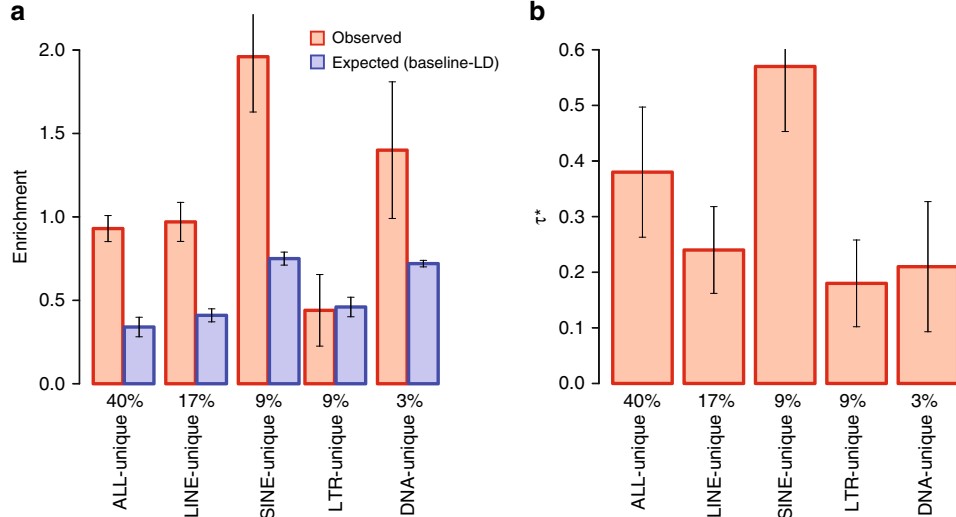

**Fig. 4** Robustness of results to difficulty in mapping to TE regions. For each of LINE-unique, SINE-unique, LTR-unique, DNA-unique, and ALLTE-unique, we report **a** two measures of heritability enrichment: observed and expected (baseline-LD); and **b** standardized effect size ($\tau^\star$), which quantifies effect that are unique to the focal annotation. Results are meta-analyzed across 41 independent traits. Numerical values of %SNPs are provided for each annotation. Error bars denote 95% confidence intervals. Numerical results are reported in Supplementary Table 21

uniquely map to. We determined that 79% of ALLTE SNPs (vs. 85% of 1000 Genomes SNPs) lie in a 35-mer that has mappability of 1 (i.e., unique mappability) based on the ENCODE 35-mer track. We refer to these SNPs as "uniquely mappable" SNPs. We first confirmed that the depletion of functional annotations from the baseline-LD model in ALLTE SNPs is still observed when restricting to uniquely mappable ALLTE (ALLTE-unique) SNPs (Supplementary Table 20), and thus cannot be explained by the difficulty in mapping to TE regions.

We repeated our heritability enrichment analysis for the main TE classes (ALLTE, LINE, SINE, LTR, and DNA; see Fig. 1) while restricting to uniquely mappable SNPs (ALLTE-unique, LINE-unique, SINE-unique, LTR-unique, and DNA-unique). We continued to observe that TE are significantly informative for

disease heritability conditional on baseline-LD model annotations (Fig. 4 and Supplementary Table 21; e.g., $\tau^\star = 0.38 \pm 0.06$ for ALLTE-unique vs. $\tau^\star = 0.31 \pm 0.05$ for ALLTE). We also observed that ALLTE-unique had higher absolute enrichment than ALLTE; this can partly be explained by the older age of TE in ALLTE-unique, as a subset of ALLTE-unique with the same distribution of age as ALLTE produced enrichment estimates closer to ALLTE (Supplementary Table 22). In the above analyses we retained nonuniquely mappable SNPs in the 1000 Genomes reference panel used by S-LDSC to compute LD scores, but we obtained similar results when removing nonuniquely mappable SNPs from the reference panel as well (Supplementary Table 23). Overall, our results confirm that our results are robust to the difficulty in mapping to TE regions.

## Discussion

We have quantified the disease heritability explained by TE, including different classes of TE. We reached three main conclusions. First, TE are uniquely informative for disease heritability, as they explain substantially more heritability than expected based on their depletion for known functional annotations. This implies that TE acquire function in ways that differ from known functional annotations. Second, we observed that older TE contribute more to disease heritability, consistent with acquiring biological function. Third, the SINE class of TE is far more enriched for blood traits than for other traits, showing that TE biology can be trait class-specific.

Our findings have several biological implications. First, our results suggest that the functional annotation of the human genome is far from complete, as the functional regions underlying the contribution of TE to disease heritability have yet to be annotated. This motivates intense efforts to identify these functional regions. We have provided a framework ($\tau^\star$ metric; Fig. 1c) to evaluate these efforts. Specifically, a $\tau^\star$ value close to 0 (conditional on a new set of functional annotations) would imply that this goal has been achieved; this can be evaluated for all TE and all traits (Fig. 1c), but is of particular interest for SINE and blood traits (Fig. 2b). Second, our TE-related annotations with conditionally significant signals (Fig. 1c) can be incorporated to improve functionally informed fine-mapping[37–39], as well as functionally informed efforts to increase association power[40–42] and polygenic prediction accuracy[43,44].

We note several limitations of our work. First, S-LDSC cannot be applied to estimate the heritability enrichment of TE families/subfamilies that span a small proportion of the genome (e.g., less than 0.4% of common SNPs; see Methods)[25]. We can instead compute the heritability enrichment that is expected based on an annotation's overlap with baseline-LD model annotations, although we caution that this quantity has a different interpretation. Second, we focused our analyses on common variants, as we used the 1000 Genomes LD reference panel, but future work could draw inferences about low-frequency variants using larger reference panels[45]. Third, SNPs lying inside TE may be difficult to identify and annotate due to the challenges of aligning TE sequences. However, analyses restricted to SNPs that lie in a 35-mer that has mappability of 1 based on the ENCODE 35-mer track ("uniquely mappable" SNPs) produced similar findings (Fig. 4 and Supplementary Tables 21–23), confirming that our results are robust to the difficulty in mapping to TE regions. Fourth, our work focuses on polymorphic SNPs, and does not quantify the contribution of polymorphic TE to disease heritability. We expect the impact of polymorphic TE on our results to be extremely small, as it has been estimated that <0.05% of TEs remain active today[46], and only TEs that have inserted within the past few hundred thousand years would remain polymorphic in humans. Indeed, we confirmed that removing the set of 14,870 human-specific TE identified by ref. [47] (which is a superset of polymorphic TE) does not significantly change our results (Supplementary Table 24). Fifth, inferences about components of heritability can potentially be biased by failure to account for LD-dependent architectures[26,48,49]. All of our analyses used the baseline-LD model, which includes 6 LD-related annotations[26]. The baseline-LD model is supported by formal model comparisons using likelihood and polygenic prediction methods, as well as analyses using a combined model incorporating alternative approaches[50]; however, there can be no guarantee that the baseline-LD model perfectly captures LD-dependent architectures. Despite these limitations, our results substantially improve our current understanding of the contribution of TE to the genetic architecture of diseases and complex traits.

## Methods

We use two metrics (heritability enrichment and standardized effect size ($\tau^\star$)) to measure the contribution of an annotation to disease and trait heritability[25,26]. We define a functional annotation as an assignment of a numeric value to each SNP. Binary annotations can have value 0 or 1 only; a binary annotation can be viewed as a subset of SNPs (the set of SNPs with annotation value 1). Continuous-valued annotations can have any real value.

**Standardized effect size ($\tau^\star$).** S-LDSC assumes that the per-SNP heritability or variance of effect size (of standardized genotype on trait) of each SNP is equal to the linear contribution of each annotation[25]

$$\text{Var}(\beta_j) = \sum_c a_{cj}\tau_c, \qquad (1)$$

where $a_{cj}$ indicates the annotation value of SNP $j$ for the annotation $c$ and $\tau_c$ is the contribution of annotation $c$ to the per-SNP heritability. S-LDSC estimates the $\tau_c$ for each annotation using the following equation

$$\text{E}[\chi_j^2] = N \sum_c \ell(j,c)\tau_c + 1, \qquad (2)$$

where $N$ is GWAS sample size and $\ell(j,c)$ is the LD score for the SNP $j$ and annotation $c$ computed from the 1000 Genome project (see URLs). We estimated $\ell(j,c)$ as $\sum_k a_{ck}r_{jk}^2$ where $r_{jk}$ is the genotypic correlation between SNPs $j$ and $k$.

Because $\tau_c$ depends on trait heritability and the size of annotation we cannot compare $\tau_c$ between different traits or annotations. Gazal et al.[26] introduced standardized effect size ($\tau^\star$) for an annotation as follows

$$\tau_{c*} = \frac{\tau_c sd(c)}{h_g^2/M_c}, \qquad (3)$$

where $sd(c)$ is the standard deviation of the annotation values, $M_c$ is total number of common SNPs used to estimate the $h_g^2$, and $h_g^2$ is the SNP heritability for each trait. In our experiments $M_c$ is equal to 5,961,159. We can compare $\tau^\star$ between different traits or annotations.

**Observed %heritability and heritability enrichment.** Observed %heritability (Observed $\%h_g^2(c)$) is the proportion of heritability causally explained by the set of common SNPs in an annotation, computed as follows

$$\text{Observed }\%h_g^2(c) = \frac{\text{Observed }h_g^2(c)}{h_g^2}, \qquad (4)$$

where

$$\text{Observed }h_g^2(c) = \sum_j a_{jc}\text{Var}(\beta_j) = \sum_j a_{jc}\left(\sum_c a_{jc}\tau_c\right). \qquad (5)$$

Expected (%SNPs) is the proportion of common SNPs that lie in an annotation. Observed heritability enrichment is defined as the Observed %heritability captured by an annotation divided by Expected (%SNPs). Observed heritability enrichment is computed as follows

$$\text{Observed heritability enrichment} = \frac{\text{Observed }\%h_g^2(c)}{\text{Expected (}\%\text{SNPs}(c))} = \frac{\frac{h_g^2(c)}{h_g^2}}{\sum_j \frac{a_{jc}}{M_c}}. \qquad (6)$$

We have previously shown that replacing the denominator with % heritability that is expected based on a MAF+LD model consisting of just the 10 MAF bins + 6 LD-related annotations from the baseline-LD model (Expected (MAF + LD)) produces results very similar to Expected (%SNPs)[50].

Both Observed heritability enrichment and $\tau^\star$ are computed conditional on set of annotations in the model (e.g., the baseline-LD model[26], which includes a broad set of coding, conserved, regulatory, MAF, and LD-related annotations). Standardized effect size ($\tau^\star$) is defined as the proportionate change in per-SNP heritability associated with an increase in the value of the annotation by one standard deviation[26]; $\tau^\star$ quantifies signals that are unique to the focal annotation after conditioning on all the annotations in the model. On the other hand, enrichment quantifies signals that are unique and/or nonunique to the focal annotation.

We computed the statistical significance of Observed heritability enrichment using block jackknife, as described in our previous studies[25,26,28] where we break the genome to 200 equal blocks. We compute the statistical significance of $\tau^\star$ by assuming that $\frac{\tau*}{se(\tau*)}$ follows a normal distribution with mean zero and variance one $\left(\frac{\tau*}{se(\tau*)} \sim N(0,1)\right)$[25,26,28].

**Expected %heritability and Expected enrichment.** We computed the Expected (baseline-LD) %heritability and Expected (baseline-LD) enrichment of an annotation by assuming that the $\tau$ of the annotation is zero. This is equivalent to applying S-LDSC to each trait using the baseline-LD model and computing the per-SNP heritability for each variant using Eq. (1). We computed the Expected (baseline-LD) %heritability of an annotation by summing the per-SNP heritability of all common SNPs that lie in the annotation and dividing by the total per-SNPs

heritability of all common SNPs. We computed Expected (baseline-LD) enrichment as Expected (baseline-LD) %heritability divided by Expected (%SNPs). We computed standard errors using block jackknife as described above.

We initially considered 34 GWAS summary association statistic data sets that are publicly available and 55 UK Biobank traits (see URLs) for which summary association statistics were computed using BOLT-LMM (see URLs; up to $N = 459$ K European-ancestry samples). We restricted our analyses to 47 data sets with $z$-scores of total SNP heritability at least 6 (Supplementary Data 1). The 47 data sets included 6 traits that were duplicated in two different data sets (genetic correlation of at least 0.9). Thus, we analyzed 41 independent diseases and complex traits ($N = 320$ K).

The meta-analyzed values of Observed heritability enrichment, Expected (baseline-LD) enrichment, and $\tau^\star$ across the 41 independent traits (47 data sets, see Supplementary Table 2) were computed using a random-effect meta-analysis, as implemented in the rmeta R package (see URLs).

**TE annotations**. We constructed two annotations for each TE where the first annotation is obtained by considering all the SNPs that fall in a TE and the second annotation is obtained by considering all the SNP in a 500 bp window of the TE. The window annotation is based on recommendation of previous work[25]. All results are obtained by conditioning over baseline-LD model. The $\tau^\star$ and enrichment reported for each TE class/family/subfamily are based on the first constructed TE annotation. We compared this enrichment estimates with the case where we compute the enrichment of an annotation conditional jointly on four extra annotations created by considering different window size of 100, 200, 500, and 1000 bp. We observed that S-LDSC results does not depend on the window size (Supplementary Fig. 14 and Supplementary Table 25).

It is uncertain whether S-LDSC estimates for annotations of size smaller than 0.4% (of common SNPs) would be reliable[25], and for this reason we did not use S-LDSC to estimate Observed heritability enrichment for these annotations, but rather only estimated the Expected (baseline-LD) enrichment based on other annotations (see above).

**S-LDSC simulations**. We set the $\tau$ for each annotation based on enrichment obtained in real data sets. Utilizing the total heritability we simulated causal trait effect sizes using a polygenetic model: $\beta \sim N(0, h_g^2/n_c)$ where $n_c$ is the number of causal SNPs. We simulated the phenotypic values under the additive model ($Y = X\beta + e$), where $X$ is the standardized genotype matrix and $e$ is the environment and measurement noise. We computed the summary statistics by performing linear regression between the phenotypic values and genotype data using PLINK software (see URLs). In our simulation, we vary the number of individuals for the traits among 2000, 20,000, and 40,000 where UK biobank genotypes[51] are used. After simulating the summary statistics, we applied S-LDSC conditional on baseline-LD model and our TE annotation. Regression SNPs in S-LDSC were obtained from the HapMap Project phase 3[52] (see URLs). These SNPs are well-imputed SNPs. SNPs with marginal association statistics larger than 80 or larger than 0.001 N and SNPs that are in the major histocompatibility complex region were excluded from all the analyses[25,26,28]. Reference SNPs were obtained using the European samples in 1000G[53]. Heritability SNPs, which are used to estimate $h_g^2$, were common variants (MAF $\geq 0.05$) in the set of reference SNPs.

**Excess overlap**. Let $A$ and $B$ indicate two annotations and $|.|$ indicate the number of SNPs in the annotation. We defined the excess overlap as follows

$$\text{Excess}(A, B) = \frac{\frac{|A \cap B|}{M}}{\frac{|A|}{M} \frac{|B|}{M}}, \tag{7}$$

where $M$ is total number of SNPs and $|A \cap B|$ indicates the set of SNPs that is shared in both annotations $A$ and $B$. We compute the standard error over our estimates using block jackknife with 200 blocks that is similar how S-LDSC computed the standard error over heritability enrichment as described in our previous studies[25,26,28].

**Tissue-specific chromatin annotations**. Blood chromatin is blood active chromatin regions by combining 27 blood cells and 6 chromatin marks (H3K27ac, H3K4me3, DNase, DNase-H3K27ac, and DNase-H3K4me3) obtained from ChromImpute[54] applied on Roadmap Epigenomics data[27]. Nonblood chromatin is nonblood active chromatin regions by combining 100 nonblood cells and 6 chromatin marks (H3K27ac, H3K4me3, DNase, DNase-H3K27ac, and DNase-H3K4me3).

Brain chromatin is brain active chromatin regions by combining 13 brain cells and 6 chromatin marks (H3K27ac, H3K4me3, DNase, DNase-H3K27ac, and DNase-H3K4me3) obtained from ChromImpute[54] applied on Roadmap Epigenomics data[27]. Nonbrain chromatin is nonbrain active chromatin regions by combining 114 nonbrain cells and 6 chromatin marks.

## URLs

For baseline-LD annotations, see https://data.broadinstitute.org/alkesgroup/LDSCORE/; TE annotations are available at https://data.broadinstitute.org/alkesgroup/LDSCORE/TE/; RepeatMasker software is available at http://www.repeatmasker.org; 1000 Genomes Project Phase 3 data is available at ftp://ftp.1000genomes.ebi.ac.uk/vol1/ftp/release/20130502; PLINK software is available at https://www.cog-genomics.org/plink2; BOLT-LMM software is available at https://data.broadinstitute.org/alkesgroup/BOLT-LMM; BOLT-LMM summary statistics for UK Biobank traits is available at https://data.broadinstitute.org/alkesgroup/UKBB; rmeta R package is available at https://cran.r-project.org/web/packages/rmeta/index.html; CIS-BP is available at http://cisbp.ccbr.utoronto.ca/. Sequence logo is generated by WebLogo 3 software which is available at http://weblogo.threeplusone.com/create.cgi.

**Reporting summary**. Further information on research design is available in the Nature Research Reporting Summary linked to this article.

## Data availability

This work used summary statistics from the UK Biobank study (http://www.ukbiobank.ac.uk/). The summary statistics for UK Biobank is available online (see URL). The TE TE annotations created in this works is available online at https://data.broadinstitute.org/alkesgroup/LDSCORE/TE/.

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

## Acknowledgements

We are grateful to Armin Schoech and Po-Ru Loh for helpful discussions. This research was funded by NIH grants U01 HG009379, R01 MH101244, R01 MH109978, and R01 MH107649. This research was conducted using the UK Biobank Resource under Application 16549. F.H. is also supported by NIH grants T32 DK110919 and F32HG009987.

## Author contributions

F.H. and A.L.P. designed the experiments. F.H. performed the experiments. Fa.H., B.V. G., J.N., O.W., S.G., C.J.J, L.O., M.L.A.H, J.E., Fe.H., and A.L.P. analyzed the data. F.H. and A.L.P. wrote the paper with assistance from all authors.

## Additional information

**Competing interests:** The authors declare no competing interests.

