## [Peer Review File · Nature Communications]

Reviewers' Comments:

Reviewer #1:

Remarks to the Author:

The authors describe an interesting analysis, assessing the role of SNPs that fall in varied classes of TEs in terms of disease associations. The analysis concludes that TEs play an important role in harboring potential disease-associated SNPs.

The conclusions are novel and potentially important. However, I have several questions:

1) The conclusion that "This implies that TE acquire function in ways that differ from known functional annotations" is not supported by data shown. Specifically, if I understand correctly, the conclusion rest on the following observation (and please clarify if this is not correct): if one adjusts for known regulatory regions/annotations (as done by baseline-LD), then there is still substantial heritability that can be explained by SNPs falling in TEs. The authors make an important point that TEs are more likely to be "difficult to map to" areas of the genome. Hence, how much of the observed results are simply explained by the fact that experimental data generated from various "seq" assays are just harder to map when they correspond to TEs (i.e., the annotations themselves are "noisy" when it comes to TEs). In the Discussion the authors attempt to partly address this, but I think more in depth analysis is needed to show the quality of annotations falling within TEs is not a major factor.

2) It's hard to understand the "depletion" results and its significance. It wasn't clear to me how can TEs be depleted for causal SNPs on one hand, and then provide an important source of functional annotations for causal SNPs on the other. Also, shouldn't the comparisons in Figure 1A be a sized-matched, LD-matched, set of SNPs as opposed to %SNPs?

3) Related to above, it's interesting that the age of TEs is an important factor in enrichment analysis. Could this explain the depletion result?

Minor points:

1) The authors argue that because substantial number of common SNPs that fall in TEs are assayed, then such SNPs "can be effectively assayed". I don't see how this conclusion follows. The SNPs can be assayed, but that doesn't necessarily tell you about error rate (i.e., effectiveness) for those SNPs.

2) You state that equation (1) defines the "variance of each SNP". What is a SNP variance? (assuming you don't mean variance of the corresponding SNP genotype) You must mean the variance of beta (log odds) associated with each SNP?

3) Definition of "Expected(%SNPs)" is not clear. Please make the following definition more clear (direct) to state that Expected(%SNPs) equals the proportion of common SNPs falling in a given annotation category (?). "Under the null model, we assume the enrichment of an annotation is one. Thus, this annotation has none significant heritability enrichment (non-enriched or non-depleted), then we expect the %heritability for an annotation to be equal to %SNPs in that annotation."

Reviewer #2:

Remarks to the Author:

Very nice paper that extends on previous work from the group looking at the genomic contributions to disease heritability but extends the work to look specifically at the contribution of transposable elements (TEs). The results are mostly descriptive but I found them quite meaningful nevertheless. There's been many independent studies that have characterized the impact of some

TE families and instances on regulatory networks and in some case disease but I'm not familiar with any work that has looked systematically at their contribution to heritability.

It's interesting to see that the authors found TEs to be uniquely informative for disease heritability, beyond just the current functional annotation, and expected that this is even more true for older TEs. I was only a bit disappointed that not more was shown to try to explain the observation linking SINEs to blood traits. Overall I was quite satisfied by the description and the approached used but did have few concerns and suggestions. See more detailed comments below.

Comments:

I found the abstract to use a lot of technical jargon and be hard to understand without knowing the previous papers.

I think it would have been good to have an additional control in Fig 1 to see whether TEs really contribute more than random random regions (e.g. in Fig 1C). What about also including a set of non-TE random sequences of the genome. Would that also have a positive standardized effect size?

In your analysis of TE subfamilies, family size could introduce some bias. On page 5, you introduce a cutoff (sup table 9) but the justification for splitting into two groups at least 0.4% or less than 0.4% was poorly justified. Can you show a graph similar to sup fig 6 but based on TE family size to show that overall there's no size bias in your analysis (and that your cutoff is reasonable)? What about very small TE families? Are your estimates reliable there?

Results in Sup Table S23-25 are interesting but I feel you are not doing them justice. Could you add a figure to show: 1) the distribution of repeat class in your list of 814 families/subfamilies (how many are SINE, LINE, LTR, DNA), 2) the breakdown of families in table S23 for blood, 3) same for S24 and 4) same for S25. This will better demonstrate what you say at the top of page 7. How do you reconcile this result with the fact that that LTR didn't show much in Fig 1 and Sup Fig 6?

The association potentially identified between blood-trait and SINE is interesting but it would have been interesting to identify something specific to SINE that could help explain this. Perhaps via a motif analysis? Are there specific motifs in the SINE families that show enrichment that could help explain this?

In your paper you didn't discuss the potential contribution of polymorphic TEs. This is probably out of scope but the impact of such polymorphisms on your computations should have been described.

Other comments:

Abstract, line 6: didn't understand on first read what you meant by "54% of SNPs, $39 \pm 2\%$ of heritability; enrichment of 0.72 ± 0.03 " can you clarify?

Abstract, line 9: "expected enrichment of 0.35 ± 0.03 ; 2.11x ratio of true vs. expected enrichment" similar comment, not clear without reading further what is 0.35 here. Also a bit unclear what you mean by "functional annotation" without knowing previous papers.

Intro, line 3: perhaps split the references (2-16) between sentence 2 and 3 or add some disease relevant references to sentence #3

Page 5, top, table S6: isn't this expected given that these TE annotations are mutually exclusive?

Page 5, middle: I think sup fig 4 and sup fig 5 were inverted in the supplements

Page 5, middle: for the analysis of Sup fig4-6. Can you comment on the baseline-LD also going up with age? Is that something similar what was observed in Jacques et al. 2013 with increased overlap with open chromatin region with age? Finally, there are significant outliers in Sup fig 6, especially for DNA elements, can you comment on those? How do you deal with small TE families here?

Page 6: "Accordingly, we estimated a particularly large for SINE for blood traits (Figure 2B and Supplementary Table 18), much larger (in absolute value) than estimates for the most informative annotations in our previous work"-> could you be more explicit here and put numbers?

Sup p25: in tables S3-5 order of DNA and LTR should stay the same

Sup p25: how come the S-LDSC scores in table S4-5 are the same?

Sup p27: seems like Sup tab 9 is already included in Sup tab 10 and could be removed

Sup p28: 4 significantly depleted families could be better highlighted so we don't have to search for them

Sup p46, sup fig 8: could you also color the bars based on repeat class?

Response to reviewers for NCOMMS-18-36472-T (Hormozdiari et al.):

Reviewer #1:

The authors describe an interesting analysis, assessing the role of SNPs that fall in varied classes of TEs in terms of disease associations. The analysis concludes that TEs play an important role in harboring potential disease-associated SNPs.

The conclusions are novel and potentially important. However, I have several questions:

We thank the reviewer for suggesting that our work is interesting, novel and potentially important.

1) The conclusion that “This implies that TE acquire function in ways that differ from known functional annotations” is not supported by data shown. Specifically, if I understand correctly, the conclusion rest on the following observation (and please clarify if this is not correct): if one adjusts for known regulatory regions/annotations (as done by baseline-LD), then there is still substantial heritability that can be explained by SNPs falling in TEs. The authors make an important point that TEs are more likely to be “difficult to map to” areas of the genome. Hence, how much of the observed results are simply explained by the fact that experimental data generated from various “seq” assays are just harder to map when they correspond to TEs (i.e., the annotations themselves are “noisy” when it comes to TEs). In the Discussion the authors attempt to partly address this, but I think more in depth analysis is needed to show the quality of annotations falling within TEs is not a major factor.

The reviewer makes a good point that it is important to address whether our results could be explained by the fact that TEs are parts of the genome that are harder to uniquely map to. As noted by the reviewer, we made an effort to address this in the Discussion section of our previously submitted manuscript, but we agree that there is high value in delving into this issue more thoroughly. We thus performed the following experiments, described in a new Results subsection titled “Robustness of results to difficulty in mapping to TE regions” (p.9) and a new main figure (Figure 4):

i. Let ALLTE denote 1000 Genomes SNPs lying inside the union of all TE, ALLTE-unique denote ALLTE SNPs lying inside a 35-mer with mappability 1 (i.e. unique mappability) based on the ENCODE 35-mer track (ALLTE-unique contains 79% of ALLTE SNPs) and ALLTE-nonunique denote other ALLTE SNPs (ALLTE-nonunique contains 21% of ALLTE SNPs). For each of the 25 main binary functional annotations X of the baseline-LD model (version 1.1), we computed 4 quantities: the % of all SNPs that lie in X ($\%X_{ALL}$), the % of ALLTE SNPs that lie in X ($\%X_{ALLTE}$), the % of ALLTE-unique SNPs that lie in X ($\%X_{ALLTE-unique}$), and the % of ALLTE-nonunique SNPs that lie in X ($\%X_{ALLTE-nonunique}$)

(Supplementary Table 29). For some annotations, $\%X_{\text{ALLTE-nonunique}}$ was much smaller than $\%X_{\text{ALLTE-unique}}$ (e.g. $\% \text{DHS-Trynka}_{\text{ALLTE-nonunique}} = 1.54\%$ vs. $\% \text{DHS-Trynka}_{\text{ALLTE-unique}} = 12.85\%$), which could be explained by the older age of TE in ALLTE-unique (see below) and/or the difficulty in mapping to TE regions. However, $\%X_{\text{ALLTE-unique}}$ was smaller than $\%X_{\text{ALL}}$ for all annotations except Repressed (an annotation that is depleted for disease heritability), implying that the bulk of the finding of $\%X_{\text{ALLTE}} < \%X_{\text{ALL}}$ is due to true biology, and not due to the difficulty in mapping to TE regions.

ii. We repeated our heritability enrichment analysis for the main TE classes (ALLTE, LINE, SINE, LTR, DNA; enrichment and tau* metrics; see Figure 1) while restricting each of these TE annotations to SNPs lying inside a 35-mer with mappability 1 (i.e. unique mappability) (ALLTE-unique, LINE-unique, SINE-unique, LTR-unique, DNA-unique). (In contrast to the analysis reported in the Discussion section of our previously submitted manuscript, we retained SNPs not lying inside a 35-mer with mappability 1 as S-LDSC reference SNPs in this analysis, allowing them to contribute to genome-wide heritability). Results are reported in Figure 4 and Supplementary Table 30. Results were similar to our main results in Figure 1 (e.g. tau* = 0.38 ± 0.06 for ALLTE-unique vs. tau* = 0.31 ± 0.05 for ALLTE). We note that ALLTE-unique had higher absolute enrichment than ALLTE, which can partly be explained by the older age of TE in ALLTE-unique (see Supplementary Table 31).

iii. We obtained similar results when removing non-unique SNPs from the reference panel as well (Supplementary Table 32). (This analysis was also reported in the previous version of the manuscript.)

We believe that the findings of Figure 4 and Supplementary Tables 29-32 confirm that our results are robust to the difficulty in mapping to TE regions. We have updated the Results section (p.9) and Discussion section (p.10) accordingly.

2) It's hard to understand the "depletion" results and its significance. It wasn't clear to me how can TEs be depleted for causal SNPs on one hand, and then provide an important source of functional annotations for causal SNPs on the other.

We recognize that it is our responsibility to provide a clear exposition. We determined that TEs are depleted for disease heritability, but less depleted than expected based on their functional annotations (Expected (baseline-LD)), such that they are uniquely informative for disease heritability despite being depleted for disease heritability. Specifically, in a model that does not include TE annotations, the contribution of TE SNPs to heritability is underestimated. We have modified the Results section (p.5) to clarify this point.

Also, shouldn't the comparisons in Figure 1A be a sized-matched, LD-matched, set of SNPs as opposed to %SNPs?

Figure 1A compares the % heritability for each annotation (Observed %h²) to both what is expected based solely on the %SNPs (Expected (%SNPs); also see response to Reviewer #1 Minor comment 3) and what is expected based on the baseline-LD model (Expected (baseline-LD)).

We believe that the first comparison (Observed %h² / Expected (%SNPs)) is an appropriate primary comparison, because it is the definition of heritability enrichment used by our previously published S-LDSC method (Finucane et al. 2015 Nat Genet). We have modified the Results section (p.3-4) to note that Observed heritability enrichment equals Observed %h² / Expected (%SNPs), adding a citation to Finucane et al. 2015 Nat Genet to make clear that this is the definition used in our previously published work. We have also clarified this in the Methods section (p.14), which we have re-organized to improve clarity. In particular, we note that the S-LDSC method operates by regressing chi-square association statistics on LD scores computed with respect to multiple overlapping functional annotations, and thus accounts for LD tagging effects. We have modified the Results section (p.3) to clarify this point.

The baseline-LD model includes MAF-dependent, LD-dependent and functional annotations, modeling the impact of all of these annotations on causal variant effect sizes. Thus, Expected (baseline-LD) accounts for MAF, LD, and functional annotations.

We have also previously shown that the % heritability of each annotation relative to what is expected based on a MAF+LD model consisting of just the 10 MAF bins + 6 LD-related annotations from the baseline-LD model (Expected (MAF+LD)) produces results very similar to Expected (%SNPs) (Gazal et al. <https://www.biorxiv.org/content/early/2018/11/19/256412>; accepted in principle, Nat Genet). We have updated the Methods section (p.14) to note this concordance.

3) Related to above, it's interesting that the age of TEs is an important factor in enrichment analysis. Could this explain the depletion result?

We agree that the relationship between age of TEs and heritability enrichment is interesting. Specifically, although all 5 age quintiles of TEs were depleted for heritability, younger quintiles were more depleted and older quintiles were less depleted (Supplementary Figure 5 and Supplementary Table 7). Given that both the result that TEs are depleted for heritability (Observed < 1) and the result that they are less depleted than expected based on their functional annotations (Observed > Expected (baseline-LD)) were consistent across quintiles, we believe that age alone cannot explain these results. We have modified the Results section (p.5-6) to clarify this point.

Minor points:

1) The authors argue that because substantial number of common SNPs that fall in TEs are assayed, then such SNPs “can be effectively assayed”. I don’t see how this conclusion follows. The SNPs can be assayed, but that doesn’t necessarily tell you about error rate (i.e., effectiveness) for those SNPs.

The reviewer makes a good point that our result that the proportion of SNPs in each TE class slightly exceeded the proportion of the genome spanned by the TE class (Supplementary Figure 1) does not confirm that SNPs lying inside TE can be effectively assayed. We have thus changed “confirms that SNPs lying inside TE can be effectively assayed” to “provides an initial indication that SNPs lying inside TE can potentially be assayed” (Results section, p.4). We have also modified this sentence to cite the new Results subsection titled “Robustness of results to difficulty in mapping to TE regions” (see response to Reviewer #1 Main comment 1).

2) You state that equation (1) defines the “variance of each SNP”. What is a SNP variance? (assuming you don’t mean variance of the corresponding SNP genotype) You must mean the variance of beta (log odds) associated with each SNP?

We agree that “variance of each SNP” is not correct, and have changed this to “variance of effect size (of standardized genotype on trait) of each SNP” (Methods section, p.13).

3) Definition of “Expected(%SNPs)” is not clear. Please make the following definition more clear (direct) to state that Expected(%SNPs) equals the proportion of common SNPs falling in a given annotation category (?). “Under the null model, we assume the enrichment of an annotation is one. Thus, this annotation has none significant heritability enrichment (non-enriched or non-depleted), then we expect the %heritability for an annotation to be equal to %SNPs in that annotation.”

We agree that the prior text was not clear, and have changed this to simply state that Expected (%SNPs) is equal to the proportion of common SNPs that lie in the annotation (Results section, p.4 and Methods section, p.14; we have reorganized the Methods section to improve clarity).

Reviewer #2 (numbers added to main reviewer comments):

Very nice paper that extends on previous work from the group looking at the genomic contributions to disease heritability but extends the work to look specifically at the contribution of transposable elements (TEs). The results are mostly descriptive but I found them quite meaningful nevertheless. There's been many independent studies that have characterized the impact of some TE families and instances on regulatory networks and in some case disease but I'm not familiar with any work that has looked systematically at their contribution to heritability.

It's interesting to see that the authors found TEs to be uniquely informative for disease heritability, beyond just the current functional annotation, and expected that this is even more true for older TEs. I was only a bit disappointed that not more was shown to try to explain the observation linking SINEs to blood traits. Overall I was quite satisfied by the description and the approach used but did have few concerns and suggestions. See more detailed comments below.

We thank the reviewer for suggesting that our results are meaningful and interesting.

Comments:

1. I found the abstract to use a lot of technical jargon and be hard to understand without knowing the previous papers.

The reviewer makes a good point that the Abstract of our previously submitted manuscript contained a lot of technical content. We have removed the technical content from the Abstract (p.1).

2. I think it would have been good to have an additional control in Fig 1 to see whether TEs really contribute more than random regions (e.g. in Fig 1C). What about also including a set of non-TE random sequences of the genome. Would that also have a positive standardized effect size?

As suggested by the reviewer, for ALLTE and the four main TE classes, we constructed annotations of the same size based on randomly selected control regions (ALLTE-RANDOM, LINE-RANDOM, SINE-RANDOM, LTR-RANDOM, DNA-RANDOM) and ran S-LDSC. As expected, we determined that enrichment estimates were not significantly different from 1 and tau* estimates were not significantly different from 0 (Supplementary Figure 3; cited in the Results section, p.5). We elected to present these results in a Supplementary Figure rather than in Figure 1 because we view them as a useful check that our published S-LDSC method is performing correctly, and not as novel findings. We are open to moving them to Figure 1 at the judgment of the reviewers and editor.

3. In your analysis of TE subfamilies, family size could introduce some bias. On page 5, you introduce a cutoff (sup table 9) but the justification for splitting into two groups at least 0.4% or less than 0.4% was poorly justified. Can you show a graph similar to sup fig 6 but based on TE family size to show that overall there's no size bias in your analysis (and that your cutoff is reasonable)? What about very small TE families? Are your estimates reliable there?

We used the annotation size cutoff of 0.4% because our previous work (Finucane et al. 2015 Nat Genet) has indicated that S-LDSC is not well-suited to analysis of very small annotations, and because 0.4% is the smallest annotation size of the annotations analyzed in Finucane et al. 2015 Nat Genet; although this choice is somewhat arbitrary, we believe it is a reasonable choice based on our published work. We have modified the Methods section (p.16) to clarify this choice, and have added a corresponding citation to the Methods section when introducing the 0.4% cutoff in the Results section (p.6).

We believe that S-LDSC estimates are reliable for annotations of size larger than our 0.4% cutoff (see above). It is uncertain whether S-LDSC estimates for annotations of size smaller than 0.4% would be reliable, and for this reason we do not use S-LDSC to estimate Observed heritability enrichment for these annotations, but rather only estimate the Expected (baseline-LD) enrichment based on other functional annotations. We have modified the Methods section (p.16) to clarify this point, and have added a corresponding citation to the Methods section when introducing the 814 TE families/subfamilies of size less than 0.4% in the Results section (p.6). We also discuss this limitation in the Discussion section (p.10), and have added a corresponding citation to the Methods section there as well.

As suggested by the reviewer, we have produced a Supplementary Figure analogous to Supplementary Figure 6 (now Supplementary Figure 7) in which we plot TE family/subfamily Expected (baseline-LD) enrichment (meta-analyzed across 41 traits) vs. annotation size, for all 814 families/subfamilies with annotation size <0.4%. Results are presented in Supplementary Figure 8 (cited in the Results section, p.6). We did not observe any substantial correlation between Expected (baseline-LD) enrichment and annotation size ($R^2 = 0.00$ for LINE, 0.00 for SINE, 0.01 for DNA and 0.10 for LTR). We note that, analogous to Supplementary Figure 6 (now Supplementary Figure 7), we used Expected (baseline-LD) enrichment in this figure because S-LDSC enrichment estimates for very small annotations may not be reliable.

4. Results in Sup Table S23-25 are interesting but I feel you are not doing them justice. Could you add a figure to show: 1) the distribution of repeat class in your list of 814 families/subfamilies (how many are SINE, LINE, LTR, DNA), 2) the breakdown of families in table S23 for blood, 3) same for S24 and 4) same for S25. This will better demonstrate what you say at the top of page 7. How do you reconcile this result with the fact that that LTR didn't show much in Fig 1 and Sup

Fig 6?

As requested by the reviewer, we have added a new Supplementary Figure 13 (cited in the Results section, p.8) reporting 1) the distribution of repeat classes for all 814 families/subfamilies spanning less than 0.4% of common SNPs, 2) the distribution of repeat classes for the subset of 27 families/subfamilies reported in Supplementary Table 23 (now Supplementary Table 25) with significantly different Expected (baseline-LD+blood chromatin) enrichment for blood vs. other traits, 3) the distribution of repeat classes for the subset of 27 families/subfamilies reported in Supplementary Table 24 (now Supplementary Table 26) with significantly different Expected (baseline-LD+blood chromatin) enrichment for blood vs. other traits, 4) the distribution of repeat classes for the subset of 109 families/subfamilies reported in Supplementary Table 25 (now Supplementary Table 27) with significantly different Expected (baseline-LD+brain chromatin) enrichment for brain-related vs. other traits. We determined that majority of TE families/subfamilies for that were specifically enriched for autoimmune diseases are endogenous retroviruses (ERV), which belong to the LTR repeat class. We have updated the Results section (p.8) to describe these results.

5. The association potentially identified between blood-trait and SINE is interesting but it would have been interesting to identify something specific to SINE that could help explain this. Perhaps via a motif analysis? Are there specific motifs in the SINE families that show enrichment that could help explain this?

We thank the reviewer for the valuable suggestion to perform a motif analysis. For each of the 281 9-mers occurring in the consensus sequence of the Alu family (which spans 80% of the SINE class), we defined a genomic annotation based on regions of the genome that match the 9-mer with at most one sequence mismatch, and analyzed the genomic annotation using S-LDSC. For one 9-mer, GCGGTGGCT, the resulting annotation (spanning 0.5% of SNPs) had much higher enrichment for blood traits (6.13 ± 1.59) than for non-blood traits (-0.64 ± 0.69) ($P = 4.64E-05$ for difference; significant after correcting for 281 hypotheses tested). We also determined that this genomic annotation had significantly higher excess overlap for blood open chromatin (1.05 ± 0.01) compared to non-blood open chromatin (0.86 ± 0.01 ; excess overlap < 1 indicates depletion), providing very strong statistical evidence that it has functionality specific to blood.

We repeated the analysis for all 27 possible 9-mers with one sequence mismatch to GCGGTGGCT (effectively testing a total of $281 * 27 = 7587$ hypotheses). For one 9-mer, GTGGTGGCT, the resulting annotation (spanning 0.6% of SNPs) had much higher enrichment for blood traits (6.65 ± 1.37) than for non-blood traits (-0.70 ± 0.58) ($P = 4.63E-07$ for difference; significant after correcting for $281 * 27 = 7587$ hypotheses tested). We also determined that this

genomic annotation had significantly higher excess overlap for blood open chromatin (0.98 ± 0.01) compared to non-blood open chromatin (0.81 ± 0.01), providing very strong statistical evidence that it has functionality specific to blood.

We queried both GCGGTGGCT and GTGGTGGCT against the CIS-BP transcription factor binding motif database (<http://cisbp.cabr.utoronto.ca/>; we used the default CIS-BP parameters), which reported a match between GCGGTGGCT and the ZNF33A transcription factor binding motif and a match between GTGGTGGCT and the ZNF354C transcription factor binding motif. These results suggest that the specific role of GCGGTGGCT and GTGGTGGCT in blood traits may be related to ZNF33A and ZNF354C binding.

We describe all of these results in a new paragraph of the Results section (p.7-8), a new Figure 3, and a new Supplementary Table 20.

6. In your paper you didn't discuss the potential contribution of polymorphic TEs. This is probably out of scope but the impact of such polymorphisms on your computations should have been described.

The reviewer makes a good point that we have not considered the impact of polymorphic TEs on our computations. We expect that the impact would be extremely small, as it has been estimated that $<0.05\%$ of TEs remain active today (Mills et al. 2007 Trends Genet), and only TEs that have inserted within the past few hundred thousand years would remain polymorphic in humans. Indeed, we confirmed that removing the set of 14,870 human-specific TE identified by Tang et al. 2018 DNA Res (which is a superset of polymorphic TE) does not significantly change our results. We have updated the Discussion section (p.10) to discuss this result, citing a new Supplementary Table 33.

Other comments:

7. Abstract, line 6: didn't understand on first read what you meant by "54% of SNPs, $39 \pm 2\%$ of heritability; enrichment of 0.72 ± 0.03 " can you clarify?

We have removed the technical content from the Abstract (p.1) (also see response to Reviewer #2 Comment 1). These results are now reported only in the Results section (p.4), after the description of the methodology provided in the Overview of methods subsection (p.3-4).

8. Abstract, line 9: "expected enrichment of 0.35 ± 0.03 ; 2.11x ratio of true vs. expected enrichment" similar comment, not clear without reading further what is 0.35 here. Also a bit unclear what you mean by "functional annotation" without knowing previous papers.

We have removed the technical content from the Abstract (p.1) (also see response to Reviewer #2 Comment 1). These results are now reported only in the Results section (p.5), after the description of the methodology provided in the Overview of methods subsection of the Results section (p.3-4). We have updated the Overview of method subsection of the Results section to provide a definition of “functional annotation” (p.3).

9. Intro, line 3: perhaps split the references (2-16) between sentence 2 and 3 or add some disease relevant references to sentence #3

As requested by the reviewer, we have added some disease-relevant references (ref. 15-20) to sentence 3 of the Introduction (p.3), which we have now merged with sentence 2.

10. Page 5, top, table S6: isn't this expected given that these TE annotations are mutually exclusive?

The reviewer makes a good point that the four main TE classes are approximately mutually exclusive, and thus it is expected that tau* estimates would remain significant when conditioning these TE annotations on each other. We have updated the Results section text (p.5) to clarify this point.

11. Page 5, middle: I think sup fig 4 and sup fig 5 were inverted in the supplements

We have fixed the order of Supplementary Figures 4 and 5.

12. Page 5, middle: for the analysis of Sup fig4-6. Can you comment on the baseline-LD also going up with age? Is that something similar what was observed in Jacques et al. 2013 with increased overlap with open chromatin region with age? Finally, there are significant outliers in Sup fig 6, especially for DNA elements, can you comment on those? How do you deal with small TE families here?

Our interpretation of the result that Expected (baseline-LD) enrichment increases with age is that older TEs contain more known functional elements. This is indeed similar to what was observed in Jacques et al. 2013 PLoS Genet, who reported increased overlap with open chromatin regions for older TEs—although the baseline-LD model includes a broad set of coding, conserved, regulatory and LD-related annotations. We have updated the Results section (p.6) to clarify these points.

The reviewer is correct that there are significant outliers in Supplementary Figure 6 (now Supplementary Figure 7), especially for DNA transposons; for example, the MER121 subfamily has a Baseline-LD (Expected) enrichment of 6.21 (s.e. 0.46). (Indeed, MER121 is one of the 46 TE families/subfamilies spanning less than 0.4% of common SNPs highlighted in Supplementary Figure 10 and Supplementary Table 14 as having significant Baseline-LD

(Expected) enrichment). This implies that the MER121 subfamily is very strongly enriched for known functional elements. Several of the outliers in Supplementary Figure 6 (now Supplementary Figure 7), including MER121, were previously reported in Jacques et al. 2013 PLoS Genet to have high overlap with open chromatin regions. We have updated the caption of Supplementary Figure 6 (now Supplementary Figure 7) to clarify these points.

We deal with small TE families/subfamilies (<0.4% of common SNPs) by using Baseline-LD (Expected) enrichment; see response to Reviewer #2 Comment 3.

13. Page 6: “Accordingly, we estimated a particularly large [tau*] for SINE for blood traits (Figure 2B and Supplementary Table 18), much larger (in absolute value) than estimates for the most informative annotations in our previous work” -> could you be more explicit here and put numbers?

We have updated the Results section (p.7) to include an explicit number for tau* for SINE for blood traits (0.88±0.14). The tau* for the most informative annotation in ref. 22 (Gazal et al. 2017 Nat Genet) (-0.46±0.02; predicted allele age), and tau* for the most informative annotation in ref. 25 (Hormozdiari et al. 2018 Nat Genet) (0.52±0.05; GTEx MaxCPP) are now reported earlier in the Results section (p.5), where we have also added an explicit number for tau* for SINE for all traits (0.43±0.06).

14. Sup p25: in tables S3-5 order of DNA and LTR should stay the same

We have modified Supplementary Tables 3-5 to ensure a consistent ordering (LINE, SINE, LTR, DNA) in all tables and figures.

15. Sup p25: how come the S-LDSC scores in table S4-5 are the same?

In Supplementary Table 4, we compare the enrichment of fine-mapped causal autoimmune disease SNPs (Farh et al. 2015 Nature, Huang et al. 2017 Nature) to our S-LDSC estimates of enrichment across 41 traits. In Supplementary Table 5, we compare the enrichment of fine-mapped causal cis-eQTL SNPs (Hormozdiari et al. 2018 Nat Genet) to our S-LDSC estimates of enrichment across 41 traits. Thus, the underlying S-LDSC estimates of enrichment being compared are the same in each case. We have modified the caption of Supplementary Table 4 and 5 to clarify these points.

16. Sup p27: seems like Sup tab 9 is already included in Sup tab 10 and could be removed

The reviewer is correct that the information in Supplementary Table 9 was already included in Supplementary Table 10. We have removed Supplementary Table 9 as suggested, and retained Supplementary Table 10 (now Supplementary Table 9).

17. Sup p28: 4 significantly depleted families could be better highlighted so we don't have to search for them.

We agree and now use bold font to denote the 4 families/subfamilies that are significantly depleted after correcting for 35 hypotheses tested. We have modified the caption of Supplementary Table 10 (now Supplementary Table 9) accordingly.

18. Sup p46, sup fig 8: could you also color the bars based on repeat class?

We have color-coded the bars in Supplementary Figure 8 (now Supplementary Figure 10) based on TE class (LINE, SINE, LTR, DNA), as requested.

Reviewers' Comments:

Reviewer #1:

Remarks to the Author:

The authors have addressed my questions and concerns.

Reviewer #2:

Remarks to the Author:

The authors have very thoroughly addressed my comments, i have no further concerns.

I know that i previously recommended making the abstract less technical but this time i find it's almost too vague. Not sure if a better middle ground could be found. In any case, i do think that in the final sentence "Our results elucidate the biological roles that TE play..." is a bit too strong and should be rephrased.